# Automatic Scheduling Tool for Balloon-Borne Planetary Optical Remote Sensing

Zhen Shi [1,2] , Yong Zhao [1,2,3] , Fei He [1,2,3,*] , Zhonghua Yao [1,2,3] , Zhaojin Rong [1,2,3] and Yong Wei [1,2,3]

1   CAS Key Laboratory of Earth and Planetary Physics, Institute of Geology and Geophysics, Chinese Academy of Sciences, Beijing 100029, China; shi_zhen@mail.iggcas.ac.cn (Z.S.); zhaoyong1@mail.iggcas.ac.cn (Y.Z.); z.yao@mail.iggcas.ac.cn (Z.Y.); rongzhaojin@mail.iggcas.ac.cn (Z.R.); weiy@mail.iggcas.ac.cn (Y.W.)
2   College of Earth and Planetary Sciences, University of Chinese Academy of Sciences, Beijing 100049, China
3   Lenghu Observatory of Planetary Science, Institute of Geology and Geophysics, Chinese Academy of Sciences, Beijing 100029, China
*   Correspondence: hefei@mail.iggcas.ac.cn

**Abstract:** The balloon-borne Planetary Atmosphere Spectroscopic Telescope (PAST), China's first planetary optical remote-sensing project, will be launched for testing and conducting scientific flights during 2021 and 2022. Images of the planetary atmosphere and plasma in ultraviolet and visible wavelengths will be used to investigate the diversity of the planetary space environment in the solar system and their different drivers. Because simultaneous observation of multiple target planets in the solar system is possible, effective observation scheduling is critical to acquire high scientific merit spectroscopic imaging data. Herein, we demonstrate an automatic scheduling tool (AST) to aid the planning of observation schedules. The AST is primarily based on a planetary ephemeris and is realized on the basis of the geometrical information and optical requirements of the telescope. The temporal variations of the planetary reference frames can also be obtained to assist in the positioning and data processing of the telescope. As a part of the Chinese deep-space exploration plan, several ground-based planetary optical telescopes will be constructed in China in the future. With the use of the proposed AST, such telescopes can achieve maximum efficiency.

**Keywords:** observation schedule; balloon-borne telescope; solar system planets observation





## 1. Introduction

The balloon-borne Planetary Atmosphere Spectroscopic Telescope (PAST) is an indispensable part of the Scientific Experiment System in Near Space (SENSE) Program [1,2]. The SENSE Program, which was initiated in 2018, is a five-year Strategic Priority Research Program of the Chinese Academy of Sciences. Because of the absorption and scattering of light by atmospheric gases, ground-based observations are limited by the atmospheric window. For example, the observation of the radiation at the ultraviolet wavelength, which is the primary characteristic radiation of the key compositions in planetary environments, is difficult by using ground-based telescopes. Turbulence in the terrestrial atmosphere also reduces the spatial resolution of optical systems. The PAST, which regularly floats above 35 km from the surface, rises above 99.5% of the atmosphere, and the percentage is estimated using the Naval Research Laboratory Mass Spectrometer Incoherent Scatter Radar Extension (NRLMSISE-00) atmospheric model [3]. At this height, the observation of the radiation at wavelengths of 200–400 nm is possible. Moreover, Rayleigh scattering near the ultraviolet wavelength can no longer exist, facilitating efficient astronomical observation. Consequently, diffraction-limited imaging can be utilized, allowing for a marked improvement in the observation resolution. Because the Rayleigh scattering region is avoided by the PAST, the observation of the inner planets, e.g., Mercury and Venus, can be conducted, and daytime observations are also possible. In addition, the PAST is equipped with a planet coronagraph that allows the telescope to observe weak radiation from the nebula

and plasma around a planet by blocking the strong planetary body radiation. Considering the aforementioned advantages, the PAST is an effective tool for the collection of systematic and global data related to planetary environments. Moreover, the investigation of planetary environment diversity and various drivers can be conducted with the help of the PAST and other similar devices. These observations will be essential for protecting human life, technical systems, and global infrastructure and will aid our understanding of the past and future of the Earth. Details of the PAST are listed in Table 1.

**Table 1.** Summary of Planetary Atmosphere Spectroscopic Telescope (PAST) parameters.

| Characteristic | Value |
| --- | --- |
| Flight Altitude | Above 35 km |
| Clear Aperture Diameter | 0.8 m |
| Operational Wavelength | 280–680 nm |
| Field of View | 15′ |

Because multiple planets can be simultaneously observed during a PAST flight, the development of an appropriate strategy for observation scheduling is essential. However, astronomical observation scheduling is an NP-hard dynamic multi-criterion scheduling issue and has been considered as a constrained multi-objective optimization problem by the scientific community for several decades [4]. A renowned automatic observation-scheduling tool called Spike, originally developed by the Space Telescope Science Institute for the National Aeronautics and Space Administration (NASA) Hubble Space Telescope (HST), was initially introduced to resolve the HST's scheduling issue, and has since been adopted by several other projects for similar use [5]. Currently, compared to a fully functional automatic observation-scheduling system such as Spike, a simplified and extensible scheduling tool satisfactorily meets the scheduling requirements of the PAST. However, in the future, an intelligent observation-scheduling tool will be required for the "Quaternity" planetary optical remote sensing system that integrates ground-based, balloon-borne, space-based, and Moon-based optical remote sensing for supporting the Chinese national deep-space exploration plan. The tool introduced in this study will be the cornerstone of an intelligent scheduling tool for the future.

The remainder of this paper is organized as follows. In Section 2, we describe the geometry calculation and algorithm for scheduling tool operation. We present three potential scheduling applications of a ground-based observatory, the PAST, and a joint observation by several observatories in Section 3. Finally, a discussion and outlook are provided in Section 4.

## 2. Algorithm Description

In this study, we designed an automatic scheduling tool (AST) to solve the scheduling issue associated with the PAST. The function of the tool will be extended to enable it to be adopted to solve the scheduling problem of the "Quaternity" planetary optical detection system. In fact, almost all modern professional observatories do not schedule their observation executions exclusively using automatic scheduling tools, and significant human intervention is required to develop observation plans and to make decisions [4]. The AST introduced in this study also requires considerably human assistance. As the first part of the "Quaternity" planetary optical detection system, the PAST is in the test and trial operation stage, and potential observations are primarily proposed by a governing team. Therefore, the current purpose of the AST is to provide the necessary geometric information and a reference observation plan to the management team to assist in preparing an observation plan. At present, the AST only provides supplementary information for scheduling to two types of observatories: ground-based and balloon-borne. Observation execution is constrained by the position of the target celestial body, the position of the telescope, and the limitations of the devices. To evaluate the data quality and prioritize observations, information derived from geometric data, such as airmass, is required. The

functioning of the AST includes the determination of the observation windows of potential targets constrained by geometry limitations, followed by prioritization of observations according to other reference information. Certainly, the final decision is made by a human. The details are as below.

### 2.1. Observation Windows

The motion of celestial bodies in the observer's frame of reference is complex, and the observability of a target depends on its relative position with respect to the observer and the observer's orientation. Moreover, other celestial bodies such as the Sun and Moon must also be considered. In this study, we used a virtual ground-based observatory (VGO) located at 40.0° N and 120.0° E, with an altitude of 4100 m, as an example. The geometric constraints of this VGO are listed in Table 2. A series of target observation windows can be calculated using these geometric constraints.

**Table 2.** Geometry constraints of an imaginary ground-based observatory.

| Geometry Parameter [1] | Constraint |
|---|---|
| Sun Elevation Angle [2] | <−18° [4] |
| Target Elevation Angle | >20° |
| Separation Angle with the Moon [3] | >45° |

[1] All parameters are taken at the same point in time. [2] The Sun elevation angle is the angle between the line from the observer to the Sun and the horizontal plane, which is measured in the vertical plane with up as positive and down is negative. The target elevation angle was defined in a similar way. [3] The separation angle with the Moon is the angle between the line from the observer to the target and the line from the observer to the Moon, which ranges from 0–180°. [4] The Sun elevation angle constraint is used to define whether the astronomical night is suitable for observation.

The observation windows were calculated mathematically as follows: Figure 1 shows the vectors required to describe the positions of the observer and target at a point in time. $\mathbf{T}_t$ can be calculated using Equation (1). It is apparent that the angle between $\mathbf{T}$ and $\mathbf{T}_t$ is the elevation angle of the target, which we note as *EL* here, and the angle between $\mathbf{T}_t$ and $\mathbf{X}$ is the azimuth angle of the target in the local coordinate system at $\mathbf{O'}$, which is denoted as *AZ* here.

$$\mathbf{T}_t = (\mathbf{Z} \times \mathbf{T}) \times \mathbf{Z} / |\mathbf{Z} \times \mathbf{T}| \tag{1}$$

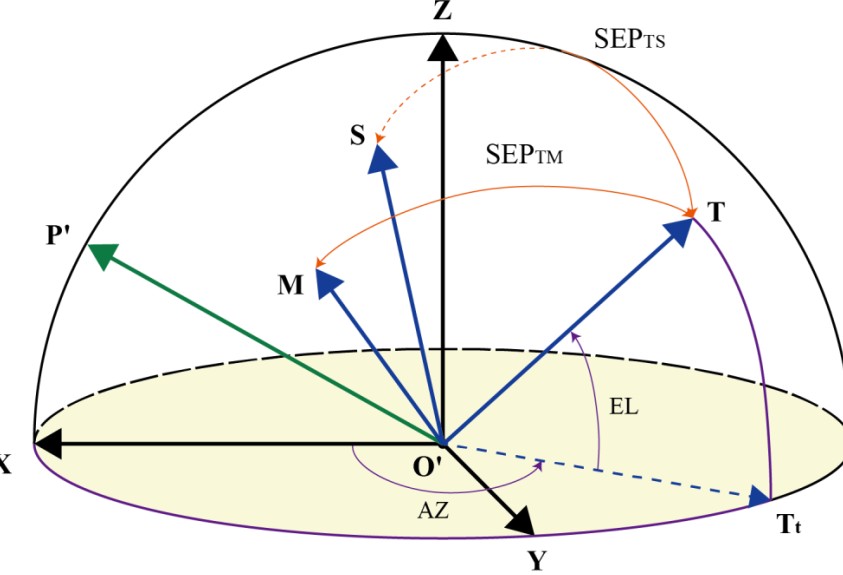

**Figure 1.** Positions of target, the Sun and Moon in local coordinate system. The observer is at $\mathbf{O'}$. All vectors are starting at $\mathbf{O'}$ and ending at a unit semi-sphere. $\mathbf{X}$ points to the north. $\mathbf{Z}$ points to zenith.

**Y** points to west. The local coordinate system is determined by these three vectors. **T** points to the target. **S** and **M** are unit vectors pointing to the Sun and Moon, respectively. $SEP_{TM}$ and $SEP_{TS}$ are separation angles between the target with the Moon and Sun, respectively. The dashed arrow, $\mathbf{T}_t$, indicates the unit vector in the direction of the projection of **T** on the horizontal plane. *AZ* and *EL* are the target azimuth and elevation angles, respectively. **P′** points to the north celestial pole.

The aforementioned vectors vary with time. However, at each point in time, we can assume that these vectors are fixed. If we regard the vectors as functions of time, *t*, we can assume that the scalars are also functions of time. Therefore, *AZ* and *EL* can be obtained through simple geometry operations. If we suppose the target is the Sun, we will obtain the Sun elevation angle, $EL_S$ (subscript *S* denotes the Sun). The separation angle with the Moon, $SEP_{TM}$, is defined as the angle between the vectors **T** and **M** (**M** starts at **O′** and ends at the Moon), as shown in Figure 1. We then substituted $EL(t)$, $EL_S(t)$, and $SEP_{TM}(t)$, into the inequality system describing the geometry constraints for ground-based observations, as in the example mentioned in Table 2, and solved it to obtain the observation windows. Theoretically, the solution to the inequality system is infinite. However, the system can be solved within a certain finite time range. For convenience, we used a Boolean function, $OBW(t)$, to describe the observation windows (see Equation (2)). The observation windows of the VGO are denoted as $OBW_V(t)$.

$$OBW(t) = \begin{cases} 1, & \text{Target is observable at } t, \\ 0, & \text{Target is unobservable at } t. \end{cases} \tag{2}$$

With respect to balloon-borne observatories such as the PAST, the geometric constraints are slightly different from ground-based observatories. For the PAST, the requirement of nighttime observations is not necessary. Instead, a constraint is required to ensure that the separation between the target and the Sun is sufficiently large. Therefore, the geometry constraints were replaced, as shown in Table 3. A significant difference between balloon-borne and ground-based observatories is that a balloon-borne observatory can move on Earth at a very high altitude. For simplicity, we assumed that the PAST was launched and made to fly at a 40-km altitude at 40.0° N and 120.0° E and then westward at a constant speed of 15 m/s. Using this simple assumption, we constructed a time position function for the PAST in the selected time range. The separation angle with the Sun, denoted as $SEP_{TS}$, is a scalar quantity that can be easily obtained when all related vectors are known. Using a similar process for the ground-based observatory, we computed the observation windows for the PAST and $OBW_P(t)$.

**Table 3.** Geometry constraints of the PAST.

| Geometry Parameter | Constraint |
|---|---|
| Separation Angle with the Sun [1] | >15° |
| Target Elevation Angle [2] | >0° & <75° [2] |
| Separation Angle with the Moon | >45° |

[1] Separation angle with the Sun is the angle between the line from the observer to the target and the line from the observer to the Sun, which is 0–180°. [2] Note that constraint 0° is an example; selection of an area below the horizon is also permitted for a balloon-borne telescope. The upper limit of <75° was selected because of the cone angle of the balloon.

### 2.2. Derived Information

In addition to the geometry constraints mentioned in Section 2.1, several data were derived from the geometric parameters that can be obtained by the AST. The derived information was applied to prioritize observations. Herein, air mass and parallactic angle are used as examples.

### 2.2.1. Air Mass

Numerous existing models can be used to calculate the airmass [6–8] (e.g., Pickering 2002, Young 1994, Young and Irvine 1967). In this study, we used a simple model, as shown in Equation (3), under a plane-parallel atmosphere assumption. This model is applicable for elevation angles of >15°. Because the constraint of the target elevation angle exceeded 20° (Table 2), this model was appropriate for use in this study. Using the observation windows and airmass, we defined a day view score (*DVS*) to evaluate the observations (see Equation (4)).

$$AM(t) = \frac{1}{\cos(\frac{\pi}{2} - EL(t))} \tag{3}$$

$$DVS(day) = \frac{100}{2.5 \times oneday} \int\limits_{1day} OBW(t)S(t)dt,$$

$$\text{Where } S(t) = \begin{cases} 3.5 - AM(t), AM(t) < 3.5 \\ 0, others \end{cases} \tag{4}$$

### 2.2.2. Parallactic Angle

As shown in Figure 2a, the parallactic angle (*PA*) is the angle between the great circle through the target and the zenith and the great circle through the target and the celestial north pole. It is also the angle between vectors $\mathbf{C}_P$ and $\mathbf{C}_Z$. The celestial sphere and parallactic angle can be calculated once vectors $\mathbf{C}_P$ and $\mathbf{C}_Z$ are known (see Figure 2b). However, the directions of $\mathbf{C}_P$ and $\mathbf{C}_Z$ are required, rather than the lengths. Using Equations (5) and (6), we can determine $\mathbf{C}_P$ and $\mathbf{C}_Z$. Moreover, *PA* is a time-dependent scalar, and this time-varying function is used to perform field de-rotation with the altitude-azimuth mounting telescope.

$$\mathbf{C}_P = (\mathbf{T} \times \mathbf{P}') \times \mathbf{T} \tag{5}$$

$$\mathbf{C}_Z = (\mathbf{T} \times \mathbf{Z}) \times \mathbf{T} \tag{6}$$

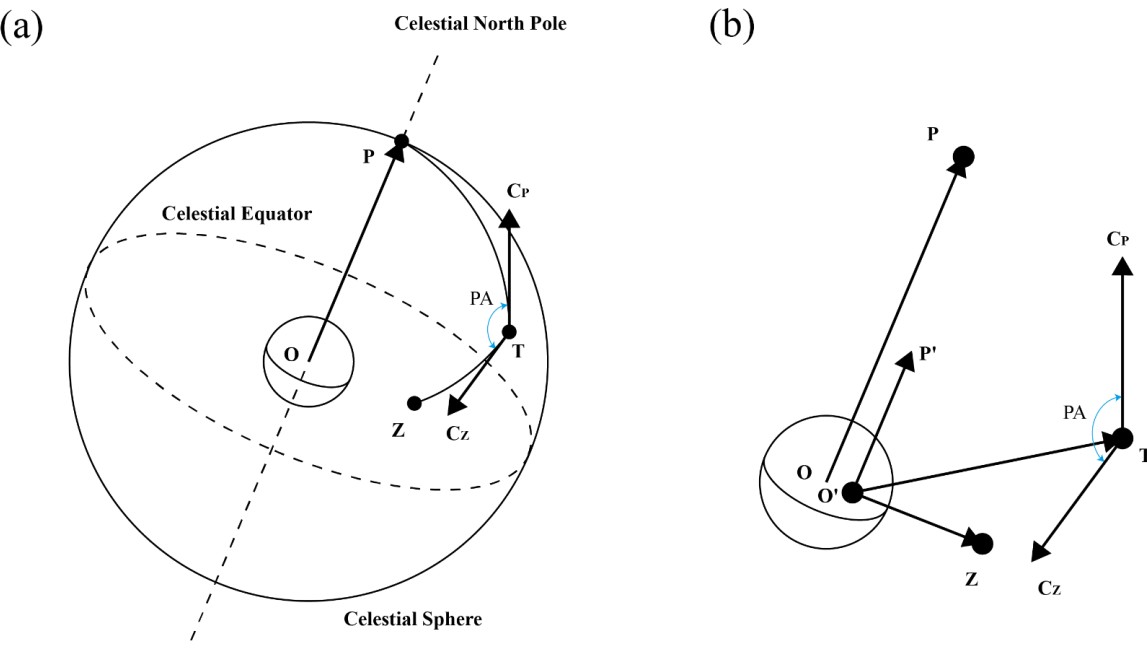

**Figure 2.** Schematic diagram of the parallactic angle. (**a**) Target and several reference points on a celestial sphere. (**b**) Vectors corresponding to (**a**). **O** is the Earth's center. **O′** is the location of the observer. **P** is a unit vector from **O** pointing to the north celestial pole. **P′** is a unit vector from **O′** pointing to the north celestial pole. $\mathbf{C}_P$ and $\mathbf{C}_Z$ are the tangent vectors of the great circle through **P** and **T** at **T** and through **Z** and **T** at **T**, respectively. The remaining points and vectors correspond to those with the same names introduced in Figure 1.

*2.3. Coding*

The current version of the AST is written in the interactive data language (IDL), and the "SPICE" [9,10] library is included to derive the positions of celestial bodies based on the DE430 ephemeris [11]. We calculated the quantities described above using time steps of 1 min. Considering the light speed, we were cautious of the difference between the apparent positions used and the true positions of the targets. Moreover, the use of the *OBW*, *DVS*, and *PA* matrices allowed us to plan observations and remediate the field rotation issue. The logical flow of the AST is summarized in Figure 3. Observation windows and derived information were obtained from the AST, and the final decision was made by humans. Finally, we obtained the observation schedule.

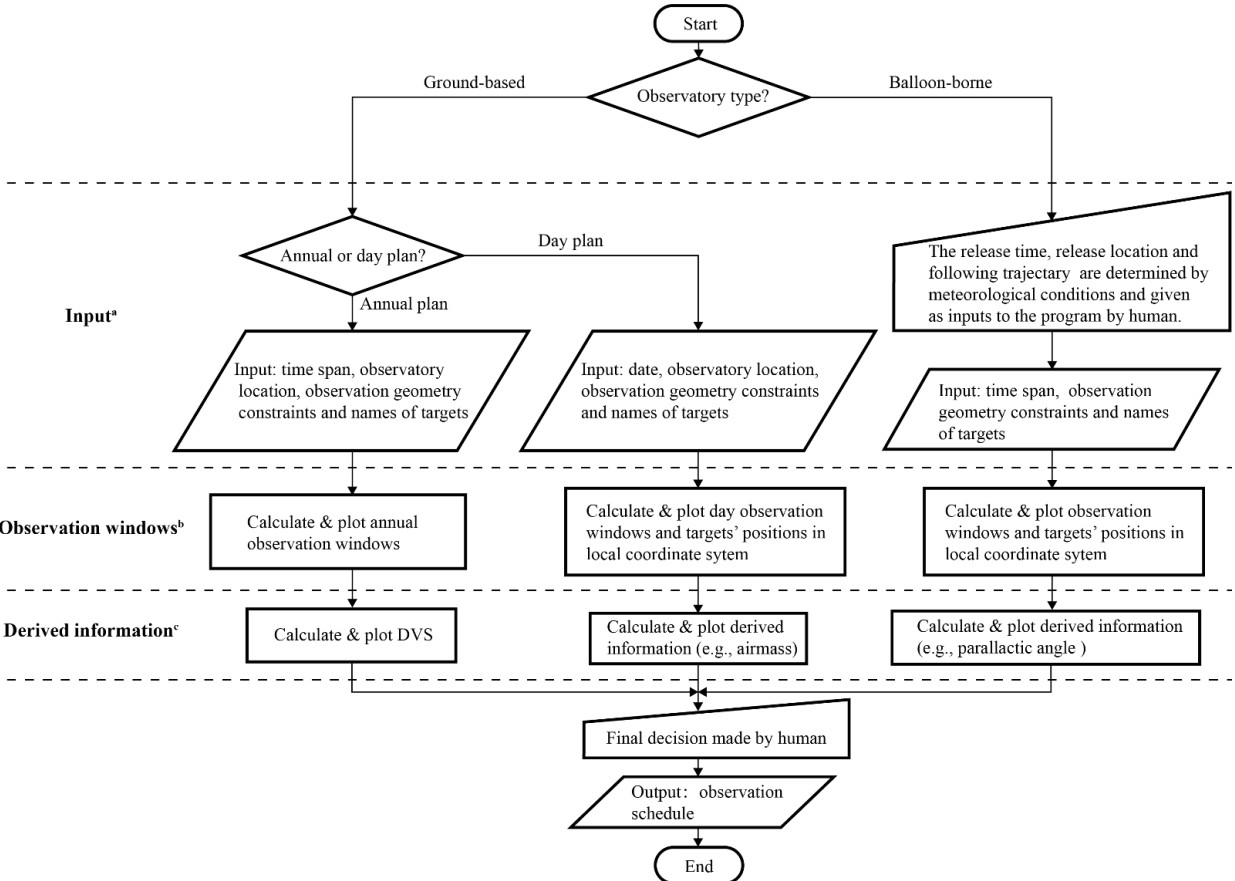

**Figure 3.** Logical flow of the automatic scheduling tool (AST): [a] The inputs of various observatory types are different. The time span required for the annual plan of a ground-based telescope is ≥1 year. The date required for a day plan is specific (e.g., 26 October 2022). The AST can only assist in the short-term scheduling for the balloon-borne telescope because the prediction of the flight path is accurate within only several days. For the ground-based telescope, the observatory location is fixed on Earth and long-term scheduling is possible. For the balloon-borne telescope, the observatory trajectory is input instead of a fixed location. Observation geometry constraints are determined by the specific observatory; [b] Observation windows are provided in days for the annual plan and minutes for the day plan or short-term plan for the balloon-borne telescope; [c] Various data can be derived from the geometry parameters for several applications.

## 3. Results

This section presents the results of the VGO and PAST using the parameters set in Section 2 at a specific time. We also demonstrate a simple application of joint observations by several observatories. Five planets were selected, namely, Mercury, Venus, Mars, Jupiter, and Saturn, as potential targets for the ground-based and balloon-borne observations. For the joint observation, the targets are introduced in Section 3.3. Note that the parameters

in the constraint can be modified, and the values used here are for demonstrating the functioning of the AST.

### 3.1. Results for Virtual Ground-Based Observatory (VGO)

First, we calculated a two-year observation window from 1 January 2021 to 31 December 2022 for the five potential targets for VGO (Figure 4); the corresponding *DVS* is shown in Figure 5. Using Figures 5 and 6, we created an annual plan. The overall frame of this annual plan is based on our observation that Mars was the only available target from 1 January 2021 to 1 May 2021; Jupiter and Saturn are available from about 1 May 2021 to 1 January 2022; and from 1 July 2022 to 31 December 2022, Mars, Jupiter, and Saturn will be observable. During the second period, Jupiter and Saturn scored closely. In the last period, Saturn was assigned the highest score, followed by Jupiter and then Mars. Moreover, the planet with the highest *DVS* can be considered as the main target; planets with higher scores are scheduled to fill up the remaining time as much as possible to maximize the active observation time and observation efficiency.

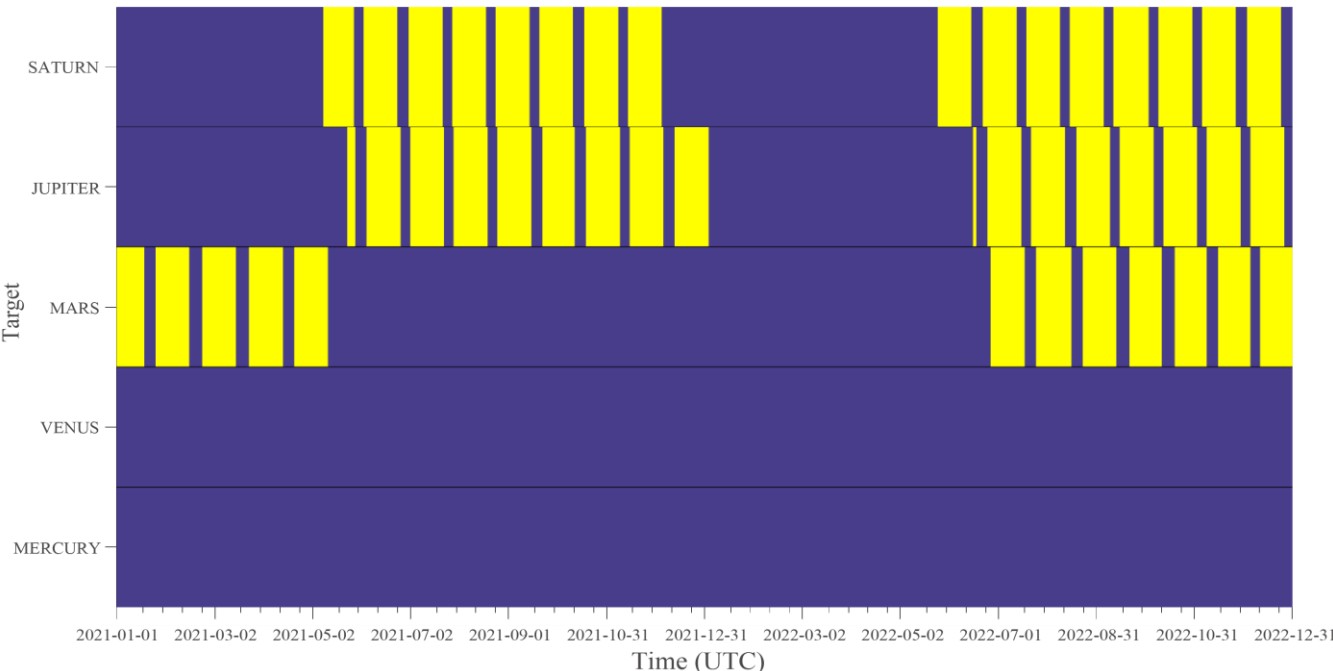

**Figure 4.** Observation windows from 1 January 2021 to 31 December 2022. Yellow areas denote the time during which the targets are observable, while dark blue areas indicate the time during which the targets cannot be observed. The horizontal axis represents UTC time and the vertical axis is the target.

In addition to establishing a long-term monitoring plan, the AST can be used to create short-term plans. In this study, we selected 26 October 2022, as an example. The observation windows for this day are shown in Figure 6. In Figure 7, the air masses of the targets are plotted. Figure 8 shows the altitude and azimuth positions of the targets in the local coordinates of the VGO. The shaded areas with specific patterns corresponding to each target indicate air masses for the observable targets. It should be noted that *DVS* was calculated as a normalized ratio of the shaded area of one day, suggesting that the longer the observation time and the lower the air mass, the higher the *DVS*. Moreover, a scheduling recommendation will be provided by the AST on 26 October 2022 (see Table 4). This scheduling dictates the observation of the target with the lowest *AM*. However, for effective observation time use, we plan to observe Jupiter from 10:37 to 15:38, and Mars from 15:39 to 20:51.

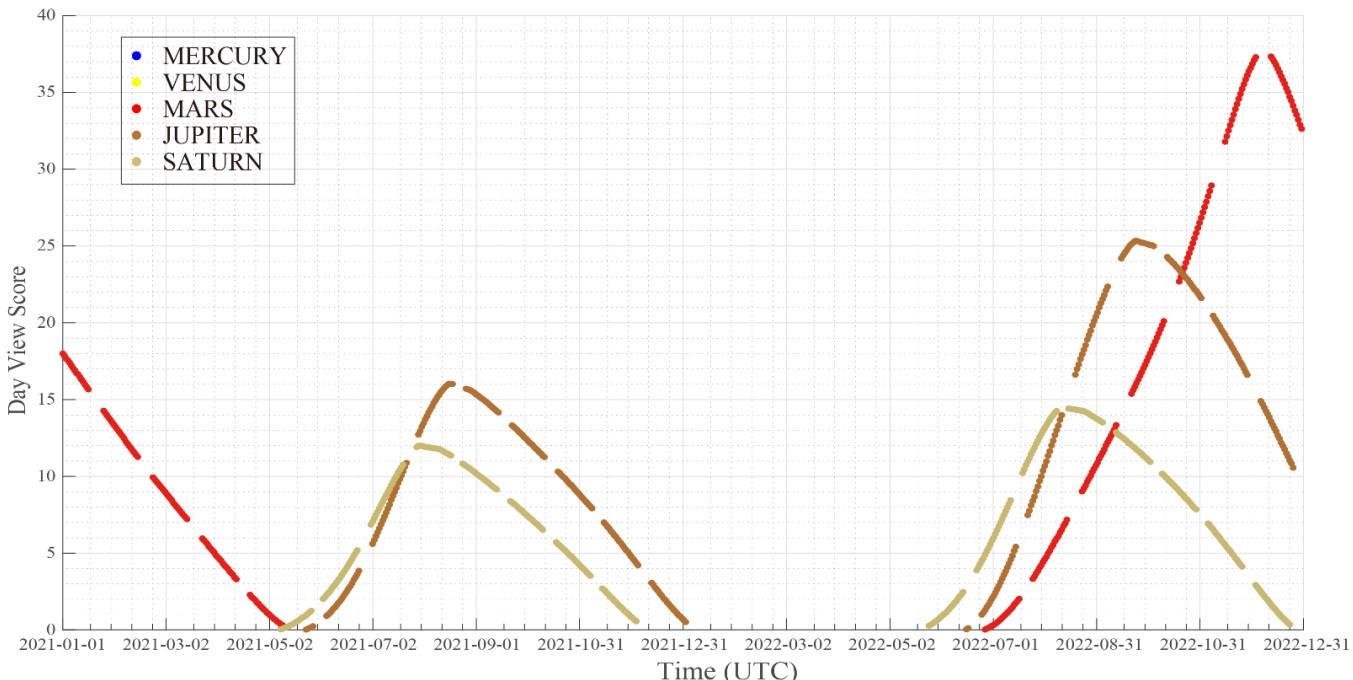

**Figure 5.** Day view score (*DVS*) of the targets from 1 January 2021 to 31 December 2022. The horizontal axis represents UTC time, and the vertical axis is the *DVS*. The *DVS* scatters shows variations of observation quality of each day during this time span. And the holes in between the points indicate unobservable time.

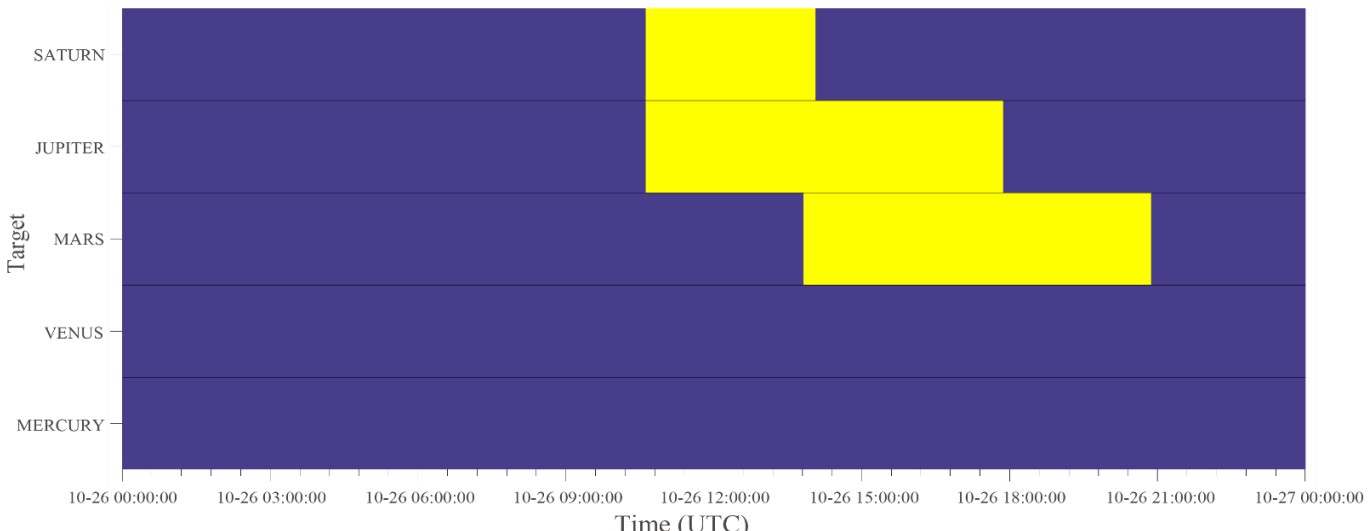

**Figure 6.** Observation windows on 26 October 2022. Information is shown in the same format as in Figure 4.

**Table 4.** Recommended schedule generated by the AST for VGO on 26 October 2022.

| Order | Target | Start Time | End Time |
|---|---|---|---|
| 1 | Saturn | 10:37 | 10:54 |
| 2 | Jupiter | 10:55 | 15:38 |
| 3 | Mars | 15:39 | 20:51 |

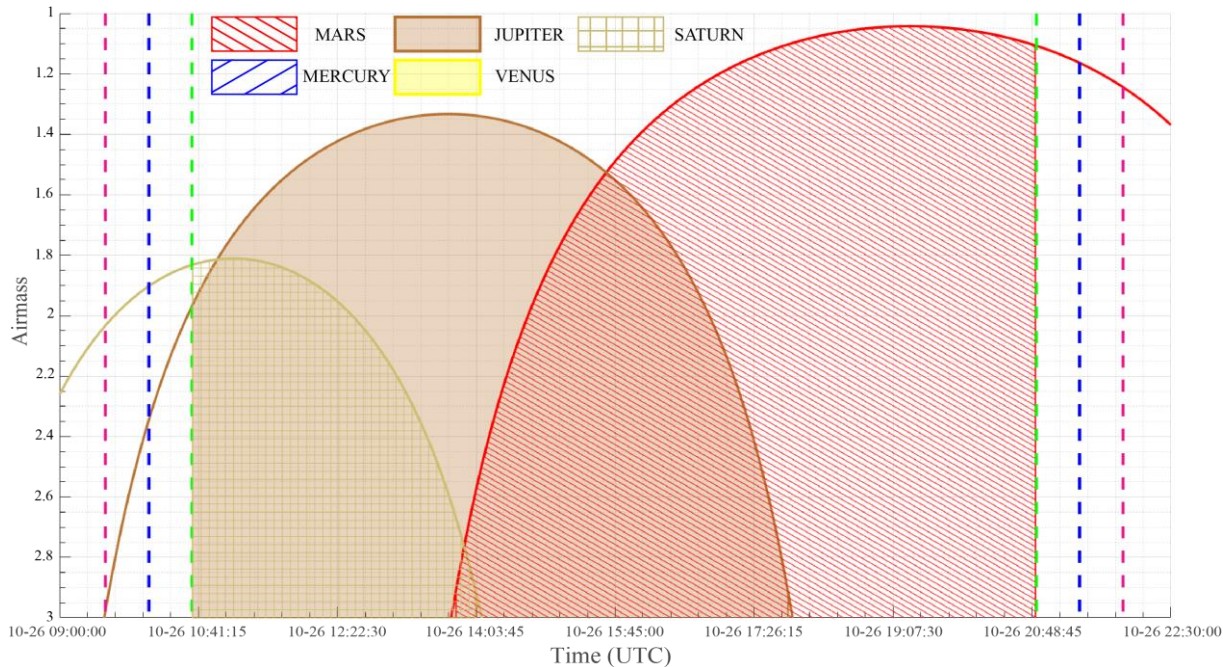

**Figure 7.** Air masses of the targets. Solid lines represent the targets' airmass. Shaded areas indicate the observable time of each target. Vertical dashed pink, blue, and green lines indicate the civil, nautical, and astronomical twilight ($-6°$, $-12°$ and $-18°$), respectively. Unobservable targets during this time span are not shown.

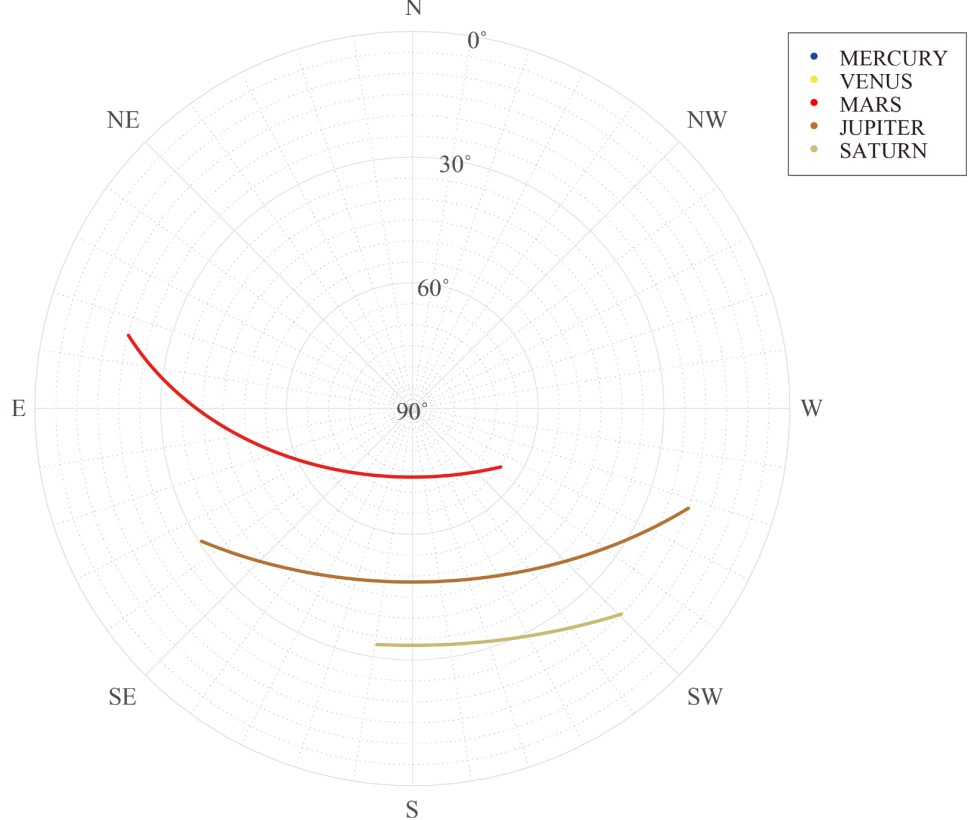

**Figure 8.** Observable altitude and azimuth positions of targets in the local virtual ground-based observatory (VGO) coordinate system at corresponding observation window (OBW) on 26 October 2022.

### 3.2. Results for Planetary Atmosphere Spectroscopic Telescope (PAST)

Because the PAST is movable and the release time and location are arbitrary, implementing an annual plan is challenging. Currently, the release time, release location, and floating trajectory are manually inputted into the AST. The release time and location are determined by meteorological conditions, such as temperature and wind speed. Once the launch window is determined and input into the AST, a schedule can be planned. The observation schedule can be dynamically updated when a new launch window is available.

The time range inspected here is from 00:00:00 on 26 October 2022 to 00:00:00 on 27 October 2022. The release location and floating trajectory are described in Section 2.1. The observation windows are shown in Figure 9. Compared to the VGO result (Figure 6) for the same day, it is apparent that the observation windows for the PAST are much broader. The corresponding target positions are shown in Figure 10. Because the aurora on planets is a field of interest for the governing team, Jupiter was selected as the first target, and the observation time was allocated to it first. Then, the remaining time was allocated to Saturn, and Mars observations have been scheduled for the remaining time.

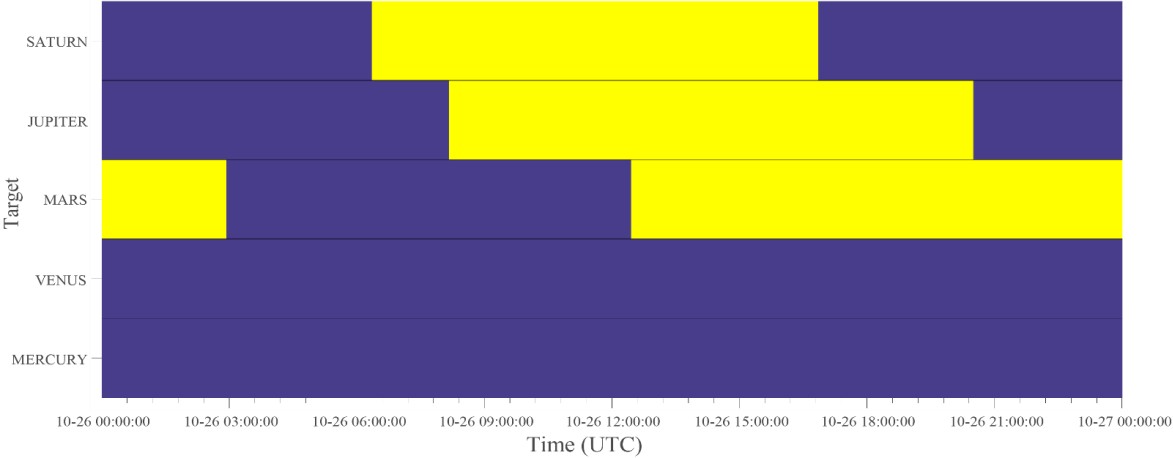

**Figure 9.** Observation windows for the PAST on 26 October 2022. Information is shown in the same format as in Figure 4.

Because the PAST is an altitude-azimuth mounting telescope, the field rotation should be compensated. The parallactic angle calculated by the AST, as shown in Figure 11, can be used to evaluate the field rotation speed. It should be noted that, on a balloon platform, the pointing of the telescope is always randomly oscillating, with an order of magnitude of 1 arcsec. The *OBW* error induced by oscillations in this study was negligible.

### 3.3. Results for Joint Observation

We applied the AST to conduct simultaneous observations of multiple targets (Mercury, Venus, Mars, Saturn, and Jupiter) from several observatories. The locations of the candidate observatories are listed in Table 5. The constraints are the same as those in Table 2, except for the PAST. For the real observation, each constraint parameter must be carefully selected according to the specific instrument. Two types of observation can be scheduled.

**Table 5.** Locations of observatories.

| Observatory | Longitude | Latitude | Altitude |
|---|---|---|---|
| VGO | 120° E | 40° N | 4100 m |
| La Silla | 70°44′ W | 29°15′ N | 2400 m |
| Mauna Kea | 155°28′15″ W | 19°49′25″ N | 4207 m |
| Xinglong | 117°34′38″ E | 40°23′45″ N | 900 m |
| Yunnan | 100°01′54″ E | 26°42′30″ N | 3200 m |
| XinJiang | 81°10′30″ E | 43°28′24″ N | 2080 m |
| PAST | Trajectory described in main text | | |

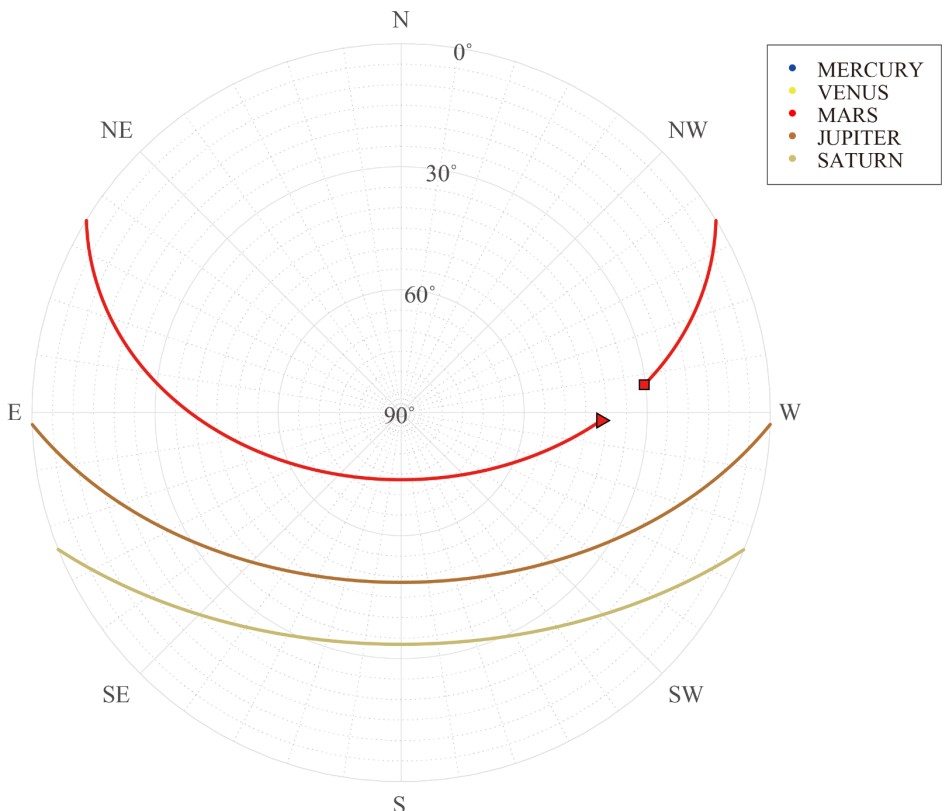

**Figure 10.** Observable altitude and azimuth positions of targets in the local PAST coordinate system at corresponding *OBW* on 26 October 2022. If the target will be observable at the beginning time and ending time of the day, the corresponding positions at these times are denoted by squares and triangles, respectively.

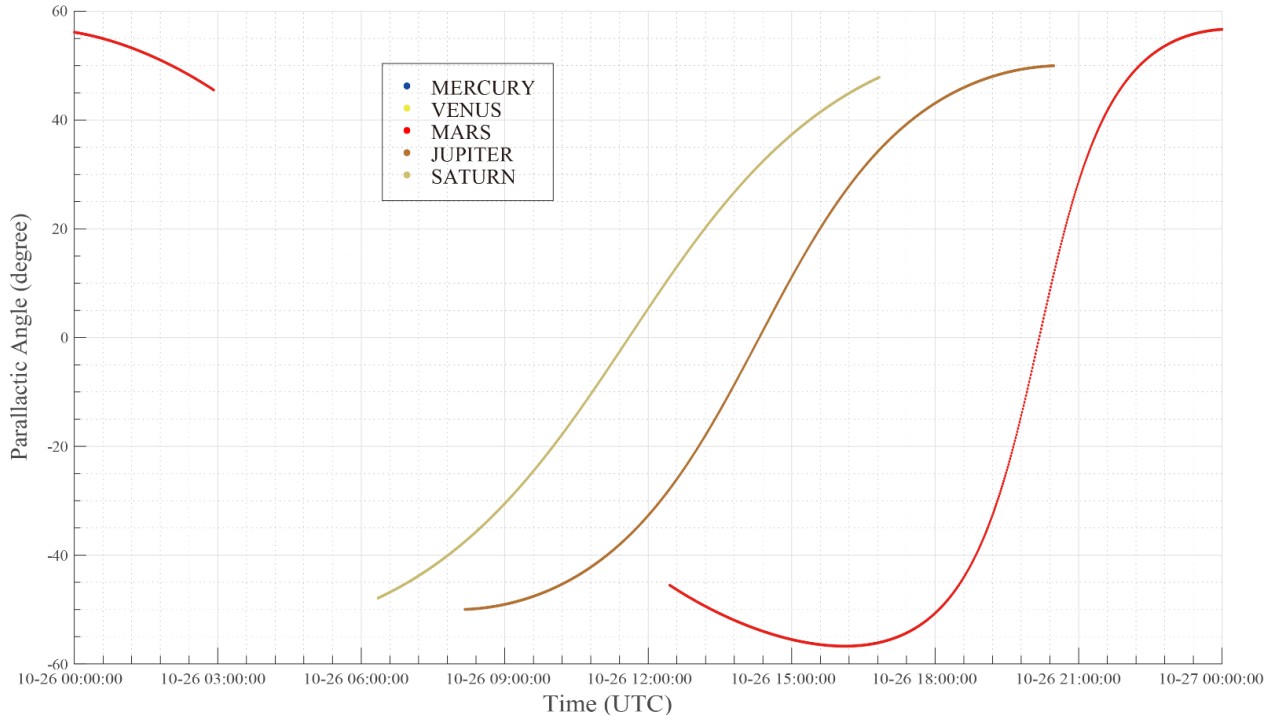

**Figure 11.** Parallactic angles of PAST targets during *OBW*. The large gap in the middle of the red line is the result of the *OBW* of Mars.

The first type is the relay observation at different observatories if equivalent instruments are available at these observatories. Taking Mars as an example, a relay observation can be carried out to maximize the observation time. As shown in Figure 12, the relay observation by La Silla, Mauna Kea, and the PAST can significantly expand the temporal coverage.

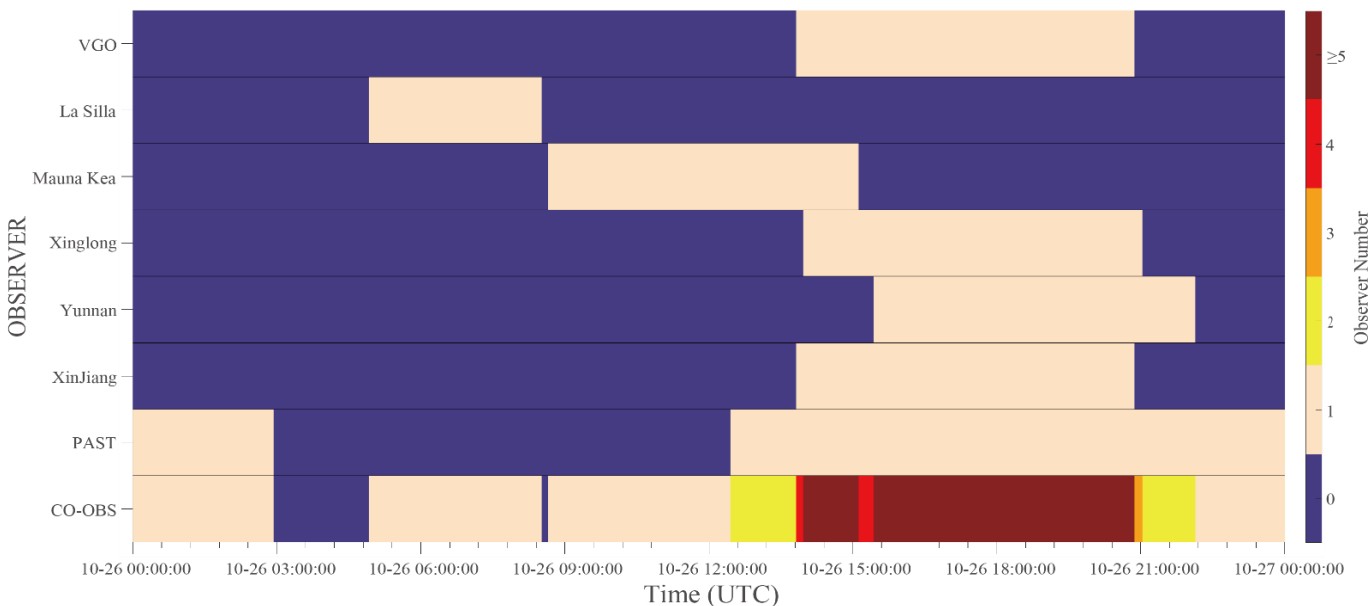

**Figure 12.** Observation windows for seven observatories and joint observation (noted as CO-OBS) for observing Mars on 26 October 2022. Colormap for single observatory indicates *OBW*. CO-OBS indicates the number of observers available simultaneously.

The second type is the simultaneous observations from different observatories using various instruments and wavelengths. This type of observation provides comprehensive information about the target. As shown in Figure 13, there are three intervals suitable for the joint observation of three planets (Mars, Saturn, and Jupiter) because the observer numbers are at least 4. For example, if Mars and Jupiter are the planets of interest, the observing plan could be observing Jupiter from 10:37:00 to 13:49:23 and then Mars until 20:50:55.

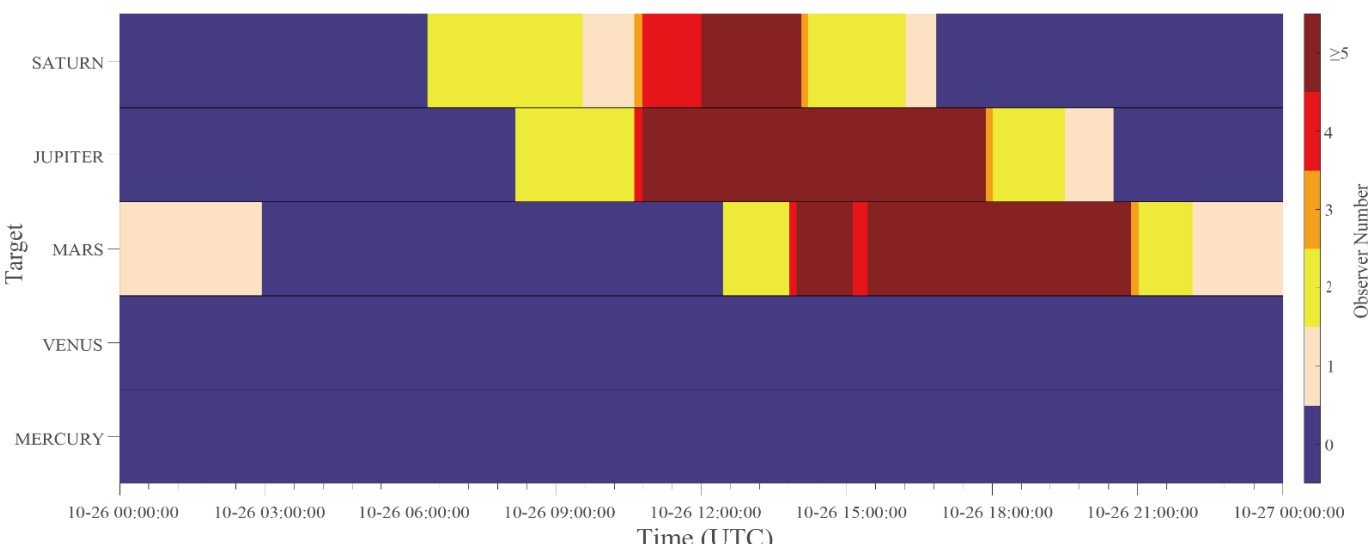

**Figure 13.** Observation windows for joint observations of five targets on 26 October 2022.

## 4. Conclusions and Future Work

The AST proposed in this study can provide observation windows based on geometry constraints and *DVS* for scheduling observations. Moreover, the AST can provide additional information, such as the parallactic angle, making it suitable for practical applications. Overall, the AST developed in this study complements the use of the PAST.

However, in the future, when the "Quaternity" observation system is established and opened to the community, the current version of the AST will not be sufficient for observation scheduling. Additional data such as the positions of celestial bodies and atmosphere and weather models will need to be introduced into the AST to improve its efficiency. Numerous optimization search methods are used by different projects to resolve the multi-objective optimization issue associated with the AST, including genetic algorithm, tabu search, and ant colony optimization [4]. Therefore, several optimization search methods will need to be implemented in the AST for its future application. Moreover, joint observations can provide a multidimensional perspective for scientists to study planetary environments, such as the combination of Jupiter's image of the PAST and Jupiter's space environment data from the Juno mission (see Appendix A). In principle, observations can be scheduled using the AST if the ephemeris of instruments beyond the Earth is known, such as the instruments on the lunar surface and the instruments encircling other planets. Furthermore, an application-programming interface (API) will be required to import data obtained from collaborative observatories. Finally, it should be noted that the AST presented here is only a preliminary version. The function of the AST will be significantly extended and updated during the construction process of the "Quaternity" planetary optical remote-sensing system in future.

**Author Contributions:** Conceptualization, Y.W., Z.Y. and F.H.; methodology, Z.S. and Y.Z.; software, Z.S.; validation, Y.Z. and F.H.; formal analysis, Z.S.; investigation, Z.S., Y.Z., Z.Y., and F.H.; resources, Z.S.; data curation, Z.S.; writing—original draft preparation, Z.S.; writing—review and editing, F.H., Z.R., Z.Y., and Y.W.; visualization, Z.S.; supervision, Z.R.; project administration, Y.W.; funding acquisition, F.H., Z.R, and Y.W. All authors have read and agreed to the published version of the manuscript.

**Funding:** This work was supported by the Strategic Priority Research Program of Chinese Academy of Sciences (Grant No. XDA17010201), the National Natural Science Foundation of China (grants 41922031 and 41774188), and the Key Research Program of the Institute of Geology & Geophysics, CAS, grant No. IGGCAS-201904.

**Data Availability Statement:** The kernels used to calculate the ephemerides of the planets are publicly available from https://naif.jpl.nasa.gov/pub/naif/generic_kernels/. The kernels used to calculate the trajectory of Juno are publicly available from https://naif.jpl.nasa.gov/pub/naif/JUNO/kernels/spk/.

**Acknowledgments:** The authors acknowledge contributions of many space scientists and the Navigation and Ancillary Information Facility (NAIF) to SPICE system.

**Conflicts of Interest:** The authors declare no conflict of interest.

## Appendix A

For scheduling cooperation with a satellite, we must know its location. Here, we present JunO's position on 7 January 2021 in Figure A1. We used the Juno Jupiter Heliospheric coordinate system in which the origin is the center of Jupiter, the X-axis is pointing to the Sun, the Z-axis is in the direction of the Sun north pole, and the Y-axis completes the system. Jupiter and its four Galilean moons are also shown.

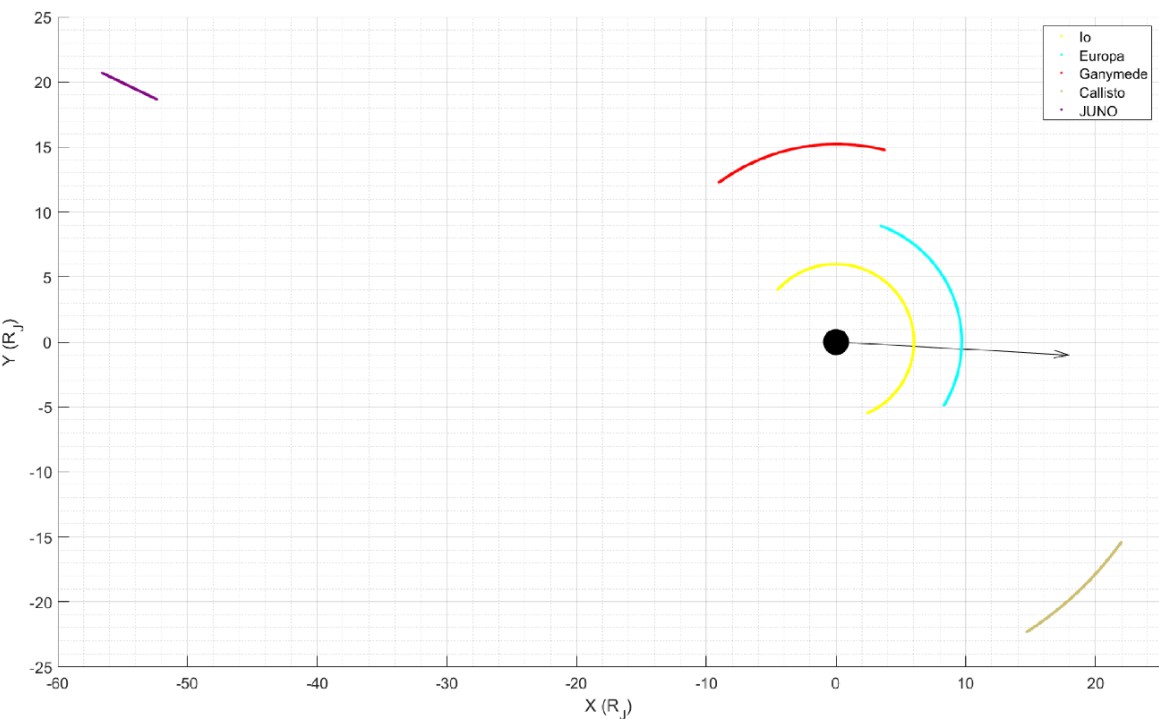

**Figure A1.** X-Y Positions of Juno, Jupiter, and its four Galilean moons. Units of both axes are the radii of Jupiter. The black sphere is Jupiter and the black arrow is pointing to the balloon-borne telescope, PAST.

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
