# Peer review of "Automatic Scheduling Tool for Balloon-Borne Planetary Optical Remote Sensing"

_remotesensing, doi:10.3390/rs13071291_

Round 1
Reviewer 1 Report
Dear Authors,
Thank you for taking into account my comments.
Please find below a few remaining comments.
general:
the Sun and the Moon should be written with capital letter then they are mentioned as astronomical objects
All the formulae seem to have broken fonts (I see a lot of rectangles with a question mark inside). The previous version was opening fine in my PC.
specific:
Table 1: "Location 1" ==> "Location" '1' is probably the footnote below, you could write directly "Longitude, latitude, altitude" in the table, even making 3 columns out of it for clarity
Fig 2: about "Apparently, P′ is parallel to P." In the previous version of this figure P' was called Pt and was just normalized P. Now P is already a unit vector, so I do not know what is the difference between P and P' (maybe it is explained in the text, but due to the problem with fonts I cannot find it).
Fig 3: about the sentence:
"Currently, the AST can only assist in the short-term scheduling for the balloon-borne telescope because the prediction of the flight path is accurate within only several days" - since as you explain the problem is not with AST, but with balloon flights, so you can drop "currently" from the sentence.
"
Fig 5: Thank you for the explanation about dots vs lines. Still I think the readers could be confused similarly to me. I would recommend to use either some kind of markers that you can see clearly that they are markers (like empty circles), or add a sentence to the legend explaining what are the holes in between the points are periods of lack of visibility due to Sun and Moon constraints.
Fig 7: the vertical lines are not well visible (especially that there is a grid in the figure, making the lines thicker would improve visibility
also in the same figure, it would be better if the objects that are not visible are removed from the legend as well. And there is a strange yellow entry without a name
Reviewer 2 Report
The manuscript contains original techniques to realize the scientific goals of a balloon-borne observing platform. Technical details of any instrumentation project are always determines how fast, and then how much science can be extracted.
Reviewer 3 Report
The revised manuscript has addressed most the issues in my former review report.
A few example language changes are provided below.
1. the sun --> the Sun
2. the moon --> the Moon
Author Response
Please see the attachment.

This manuscript is a resubmission of an earlier submission. The following is a list of the peer review reports and author responses from that submission.
Round 1
Reviewer 1 Report
Figure 3a has typo. "Celestial Equation" should read "Celestial Equator"
Reviewer 2 Report
Authors described the automatic scheduling tool (which in fact is rather half-automatic, since it only provides a set of plots for the person who does the actual schedulling). While the work done by the authors is very important for the experiment, and with the limited number of targets and the current state of the experiment it is sufficient, the software is mainly extracting planetary positions from a known software and computing a few quantities using rather simple geometrical relations. The only point in the paper which I could call in any sense novel is the DVS parameter, of which applicability however I have doubts (see below). It is clear that there is still a lot of work needed to be done (which seems to be as well the main conclusion of the authors), and solution of some of those problems will definitely require novel approaches. However, at the present the manuscript gives more impression of a work progress report rather fully finished article.
I would suggest significant expanding of the article (and corresponding software) before the publication, to tackle in more detail some of the posed problem.
A few ideas:
a) simultanuity of the observations between different observatories to either maximize the number of different instruments observing simultanously the same target, or in contrary to assure that having a few equivalent instruments the total fraction of time each of the targets is seen by at least one is maximized.
b) automatization of the (tentative) schedule, e.g. selecting automatically the source which has the lowest AM among the visible targets
c) application for instruments located beyond Earth (the Moon, JUNO, ...)
I have also a few methodology and multiple language comments below:
L15: from the rest of the paper it seems that you are interested in solar system planets. I think it should be stated explicitely in the abstract (to distinguish from exoplanet studies)
L37: "floats above 37 km, leaves 99.9% atmosphere below it"
where does the number 99.9% come from? The atmospheric depth at 37 km is of the order of 4 g / cm^2 (with some ~10% dependence on the atmospheric model used), which is only ~0.4% of the total atmospheric depth
Moreover Table 1 says 35 km instead of 37 km.
L41 "is optimized" ==> "is improved"
L55: since you mention SPIKE that was made for HST, however it is being used also in other projects, I think it would be important to explain why it cannot be simply used for PAST as well? what features you need (and implement in AST) that are not in SPIKE?
L58:
"At current stage, it’s going too far to construct a fully functional
automatic observation scheduling system like SPIKE for PAST. It’s enough to build a sim-
plified scheduling tool which is present in the following sections for current use."
I suggest rewording:
"Currently, a simplified scheduling tool (described in the following sections) is sufficient for the needs of PAST, rather than a fully functional automatic observation scheduling system like SPIKE."
L70: "will be given" ==> "is given"
(to keep the same tense as in the previous sentences)
L73: equipped ==> is equipped with
can enable ==> "enables" or "allows"
the whole footnote I think fits more as a sentence in the main text rather than a footnote in this table.
L88: "...for making final plan temporally" - the end of the sentence is not clear, maybe:
"Therefore, the current purpose ... to the management team to assist in preparing the observational plan."
L94: " is calculating" ==> "is to calculate"
L99: "Celestial bodies including Earth are always in motion." - you can remove this sentence, it is pretty obvious, and also not accurate (since motion has to be always considered w.r.t. some reference frame). Alternatively you could write something like: "The motion of celestial bodies in the observer's frame of reference is complicated."
L99: " And the observability of a target is dependent on the relative positions of target, observer... " -"And the observability of a target is dependent on its relative position with respect to the observer, and the observer's orientation. Some other celestial bodies (the Sun, the Moon) must be also taken into account. "
L101: "Those positions’ relationship is primarily a position pattern satis-
fying a certain geometry constraint of those positions." - please reconsider this sentence, I'm not sure what you want to say in here.
Table 2:
L109,110 - both footnote 2 and 3 describe the elevation angle, just the source is different, I think you can safely remove the 3rd footnote.
L113 - in some applications it is worth to constrain also moon angles close to 180 deg, i.e. if the telescope is constructed such that light detector is located at the optical axis, opposite to a mirror, then for moon angles close to 0 deg (that you constrain) the moon illuminates the mirror and thus also the light detector, however for angles close to 180 deg, if the observations are performed close to the horizon it can shine directly into the light detector.
You are also missing a condition about the position of the moon. You can imagine a case that the source is obseved 25 deg above the horizon, and the moon is 40 deg from it, but being ~-15 deg below the horizon, so you do not care about the moon light even if the angular separation between the moon and the source is not that large.
L115 astronomical twilight is defined as -18 deg. Of course depending on the application the observations might start a bit earlier, but then maybe some explanation why -15deg is sufficient would be good to be added
L119-126: those descriptions should be only included in the caption of Fig1 (and most are there already, so there is no reason to repeat it here)
Eq 1: you write that Pt and Tt are unit vectors, but from the formula they will have the length of sin (Z P angle)
L146: " Vector calculation is independent on coordinate,
which means the scalars we derived by vector calculations are identity no matter what the coordinate system we use. But when we do the computation of vectors through their components, we must make sure that the vectors participated in the process are in the same coordinate system. " - this can be removed, since it is plain math
L151: " The angle between two vectors is a
scalar which can be easily calculated when the two vectors are known" - also can be removed
L161 and Eq 3: the description says exactly the same as formula (even using the same words), you can remove either the explanation or the formula
Fig 2: it would make it much more clear to the reader if the projected rectangular of the Tg plane had its sides parallel to X and Y direction axes. You might consider to make dotted lines of T Ts Ts vectors to Tg plane (however I'm not sure if the plot does not get too crowded)
L178: why you do not need nighttime condition for balloon-born experiments? the light of the sun would still affect it, no? Or the flights are always done close to the Earth poles? This is not what you assume in L184 where you give 40^o N latitude
Table 3:
Naively thinking if you are in a baloon at 37 km you can look also a bit below the horizon, so why 0 deg elevation limit?
Also I would expect that the balloon itself would would obscure the visibility for elevation angles close to 90 deg.
L192-194 - you do not need those footnote since they are the same as in previous table, and also in the last footnote there is a mistake with Moon instead of Sun
L206: "up to about 60° to 75°" - it is not clear up to which zenith angle it is really applicable. Obviously the higher the zenith the larger the difference, so you could quote the differences at zenith 60 deg and at 70 deg (since you apply a cut in elevation above 20 deg). Actually, since you use elevation throughout the text it is better that you convert this sentence from zenith to elevation.
Eq 4
cos ==> \cos
Eq 5:
oneday ==> \mathrm{1 day}
you can remove the part when you ask AM to be >=1, since this always happens by definition.
Why 3.5 has been used? It corresponds to elevation of 17 deg, which is already below the limit. Woudn't 2.92 be more appropriate?
What is the purpose of determining the DVS? Why I see the logic in computing the airmass since it is important to quantify the sensitivity of the observations, you are not going to observe a single planet for a full day (not to mention that large part of the window will be cut by the observation conditions).
L223 Remind that ==> Note that
L239: The rest points ==> The remaining points
L242: you should explain more about what kind of code you are using here, what programing language, if you use any external libraries, the only thing that is mentioned is SPICE.
L243 -247 the first 5 sentences of the section can be summarized with simple:
"We calculate the above described quantities in time steps of 1 min."
L248 DE430 ephemeris should also have some reference given
also if you take the position of the planetes from SPICE, can't the actual observation angles be extracted directly instead of redoing the calculations? I would assume that such scheduling tools should have those information available already.
L258 why do you say that "Since AST can only assist in short term scheduling" if you put in the flow of the program information that it can assist with annual sheduling plan?
Fig 6:
The unobservable targets should not be listed in the legend if they are not plotted
in addition to the lines that give the DVS, there are also some dots that are not explained in the caption, and the legend actually shows dots instead of lines
L271 & L319 Results of VGO/PAST ==> Results for VGO/PAST
L330: for a ground based telescope it is relatively easy to calibrate the azimuth and elevation angles (i.e. to make a pointing model). I would naively expect that for a balloon experiment this is much more difficult, the "ground" plane can move around slightly due to small movements and oscillation of the balloon, I suspect that there will be some calibration constants that would affect the pointing directions and field rotation, is there a place in your software for them?
Fig 11 and Fig12: the gap in Fig 12 seems to be much larger then in Fig 11, which of the observability conditions makes the gap?
L356: you use two different formats of a references thoughout the text numbers or Author Year style, please harmonize
Reviewer 3 Report
The paper presents a methodology of balloon-borne telescope control protocol that is specially designed for PAST, the first one in China in the field of planetary studies. Given that this work is going to serve well defined and very interesting science goals, the authors came up with a plan that is convincingly practical. I found this work important, and I recommend it for publication in the journal when some moderate modifications.
- the language needs to be largely improved to help readers to understand the idea, in particular the techniques presented the paper.
- in section 2.1, the second paragraph is about the definition of the coordinate systems used by PAST. It seems to me that the authors could have made it too sophisticated by working with mixed geocentric system and the local horizontal system at the platform. May it be better to formulate the problem all in the local system and then convert it back to geocentric system? The later one is presumably the final frame of reference used in the whole work. This is just the way of presentation, which will not change the math, calculation and results.
- Balloon-borne platform astronomical observations are of course weather independent. but the launch of any balloons need to be started from the ground. I confess that I have no experience of such experiments, therefore I would like to ask the authors, as a general reader, to specify such a concern. Because the aim of this paper is PAST observation scheduling, anything that will intervene the floating of balloon, and then the science programs, should be integrated into the scheduling.
Reviewer 4 Report
Concerns, questions and clairifications:
- Table 2, in the “Separation Angle with the Moon”, lunar phase should be taken into account.
- Also in table 2, note 5, usually the astronomical observing night is considered between the end of astronomical twilight in the evening and the beginning of astronomical twilight in the next morning, and astronomical twilight is defined as the geometric center of the Sun's disk is 18 degrees below the horizon. In this case, 18 deg. is more appropriate than 15 deg.
- Table 3, “Separation Angle with the Moon” >45deg. is way too large.
- Also in Table 3, note 4 is in wrong position.
- In Section 3, the authors choose 5 planets for calculations, but only 3 superior planets (Mars, Jupiter, and Saturn) are shown in Fig. 5 - Fig. 12. The reason need to be explained.
- Time span need to be provided in Fig. 9 and Fig. 11.